# Nano-Integrated Plant Tissue Culture to Increase the Rate of Callus Induction, Growth, and Curcuminoid Production in *Curcuma longa*

**DOI:** 10.3390/plants13131819

**Published:** 2024-07-02

**Authors:** Muhammad Iqbal, Zill-e-Huma Aftab, Tehmina Anjum, Humaira Rizwana, Waheed Akram, Arusa Aftab, Zahoor Ahmad Sajid, Guihua Li

**Affiliations:** 1Department of Plant Pathology, Faculty of Agricultural Sciences, University of the Punjab, Lahore 54590, Pakistan; iqbalamaan25@gmail.com (M.I.); anjum.dpp@pu.edu.pk (T.A.); waheedakram.fas@pu.edu.pk (W.A.); 2Department of Botany and Microbiology, College of Science, King Saud University, Riyadh 11495, Saudi Arabia; hrizwana@ksu.edu.sa; 3Guangdong Key Laboratory for New Technology Research of Vegetables, Vegetable Research Institute, Guangdong Academy of Agricultural Sciences, Guangzhou 510640, China; liguihua@gdaas.cn; 4Department of Botany, Lahore College for Women University, Lahore 54000, Pakistan; arusakashif@gmail.com; 5Institute of Botany, Faculty of Life Sciences, University of the Punjab, Lahore 54590, Pakistan; zahoor.botany@pu.edu.pk

**Keywords:** nanotechnology, turmeric, curcuminoids, Fe_3_O nanoparticles, antimicrobial

## Abstract

Turmeric has attracted a significant amount of interest in recent years due to its strong antimicrobial properties. The tissue culture of turmeric is preferred to obtain disease-free, highest number of plantlets with good uniform chemistry. However, there is a need to increase the speed of the whole process to meet the growing demand for planting materials and to save time and resources. Iron oxide nanoparticles (Fe_3_O_4_ NPs) showed positive effects on callus initiation time, proliferation rate, percent root response, shoot length, percent rooting, and number of roots per explant. Highest callus induction, i.e., 80%, was recorded in cultures that were grown in the presence of 15 mg/L of Fe_3_O_4_ NPs. Callus initiated earlier in culture tubes that received green synthesized iron nanoparticles in a concentration between 10–15 mg/L. Biofabricated nanoparticles were characterized for their size, physiochemical, and optical properties through UV–Vis spectroscopy, FTIR, XRD, and SEM. Curcuminoids profiling was performed by implementing LC-Ms that revealed increased quantities in plantlets grown in nano-supplemented media when compared to the control.

## 1. Introduction

Turmeric is the common name of *Curcuma longa* L., which belongs to the family Zingiberaceae. *Curcuma longa* is a perennial herbaceous plant. This genus has 70 species of rhizomatous plants that are extensively grown throughout Asia, i.e., Pakistan, China, India, and other tropical areas [1]. Turmeric rhizomes have traditionally been utilized as antibacterial, antifungal, antiviral, insecticides, and for skin problems, wound healing, and as an anti-inflammatory agent [2]. The most valuable portion of turmeric is the rhizome, which is renowned for having antimicrobial properties. *Curcuma longa* extract is an oleoresin, made up of a substantial percentage of yellow-brown pigments and a light component of volatile oil [3]. Curcuminoids are most prevalently found in turmeric, which are naturally phenolic substances. Curcuminoids mainly consist of curcumin, demethoxycurcumin, and bisdemethoxycurcumin. Among these compounds, curcumin is the primary component of curcuminoids, which are accountable for the antimicrobial properties of turmeric. Curcumin is a naturally yellow-orange crystalline compound that is not soluble in water but is soluble in organic solvents like methanol, ethanol, and acetone. It was initially identified in 1910 as a minimal molecular weight compound with a melting point of 183 °C [4]. It has been demonstrated that turmeric extracts are highly effective at slowing the growth of various pathogenic bacteria. Ref. [5] stated that the phenolic chemicals called curcuminoids found in turmeric are what give it its antibacterial properties against *B. subtilus*, *S. aureus*, and *E. coli*. Turmeric oil demonstrated efficacious action against seven different fungi that cause agricultural products to decay when they are kept [6]. According to reports, curcumin has antifungal properties; for instance, it effectively combats human pathogenic fungi like *Paracoccidioides brasiliensis* and *Candida* spp. [7]. Curcumin has also shown high to average antifungal effects against plant pathogenic fungi including *Erysiphe graminis*, *Puccinia recondita*, *Phytophthora infestans*, and *Rhizoctonia solani*, which are causing powdery mildew in barley, wheat leaf rust, late blight of tomato, Botrytis blight in cucumber, and sheath blight of rice [8]. The extracts of *Curcuma longa*, *C. zedoaria*, *C. amada*, and *C. aromatica* successfully inhibited the growth of many fungi, i.e., *Aspergillus niger*, *A. terreus,* and *Curvularia pallescens* [9]. Due to their anti-inflammatory and therapeutic properties, curcuminoids, especially curcumin, have gained special importance. 

Turmeric rarely flowers, and being a sterile triploid plant generally reproduced by vegetative propagation of rhizomes. It is reported that out of almost 10–25 lateral buds produced by each rhizome during one growing season, only 4–6 can develop plantlets in field conditions. Accumulation of curcuminoids in rhizomes is quite slow throughout the growing season [10]. In addition to genotypic differences, variation in field and processing conditions results in differences in the chemical composition of commercial turmeric rhizomes. Tissue culture can be an option to escape all these problems. Fast multiplication of turmeric by micropropagation is required to create a continuous source of evenly sized, high-quality, and disease-free plantlets, especially for commercial purposes. Variation in responses has been observed in plants, including turmeric, with variation in media used for micropropagation [11,12]. Several reports confirm the positive role of nanomaterials in micropropagation of plants [13].

Nanotechnology is a novel field with innovative uses in plant biotechnology and agriculture. Natural substances are found in plant extracts, including flavonoids, amino acids, saponins, vitamins, terpenes, phenolics, tannins, inositols, and resins [14]. Several plant sources with these chemicals have been explored to synthesize nanoparticles with reduced involvement of synthetic chemicals, hence reducing their toxicity. Green synthesized NPs increase crop yield, hasten the germination of plants and regeneration, and strengthen plant resistance to a range of biotic and abiotic diseases [15]. Fe_3_O_4_ NPs are one of the most significant oxides that have several uses in material engineering, biological remediation, medicine, cosmetics, and agriculture [16]. Fe_3_O_4_ NPs have prospective uses as agricultural fertilizer because of their capacity to produce chlorophyll, their redox activity, magnetic sensitivity, environmentally friendly nature, and ease of availability [17]. The industrial application of apple peel extract has been reported seldom, despite the fact that apples are among the main fruits used in the manufacturing of juices, sauces, and canned fruits. The apple peels are richer in polyphenolic and flavonoid compounds than pulp. Being a great source of antioxidants and reducing compounds, apple peels may be used as an attractive substitute for the green synthesis of metal nanoparticles. Thus far, a few studies have been reported on the biogenic synthesis of nanoparticles from fruit peels. They have synthesized a range of nanoparticles from single-extract mixtures of multiple fruit peels, including orange, pomegranate, apple, and banana [18,19]. Iron is essential for plant development and functioning, the synthesis of proteins, auxin control, metabolism of carbohydrates, and stress response [20]. Fe_3_O_4_ NPs have greatly increased the regeneration and growth of chickpea, sorghum, tomato, maize, strawberry, and rapeseed crops [21,22,23,24]. The best that we are aware of is that there has not been any research conducted regarding the effect of green synthesized Fe_3_O_4_ NPs in tissue culture investigations of *Curcuma longa*. 

In this study, we present a unique green manufacturing process for apple peel extract-based Fe_3_O_4_ NPs. Furthermore, the effects of biosynthesized Fe_3_O_4_ NPs have been assessed for enhancing callogenesis, shoot regeneration patterns, root induction, and curcuminoid contents in turmeric plants. The objective of this enhancement was to increase the efficacy of micropropagation to save time and resources involved. 

## 2. Materials and Methods

### 2.1. Green Synthesis of Fe_3_O_4_ Nanoparticles

Fresh apples (variety: Kala kulu) were purchased from a local market in Lahore. The apples were washed and peeled. Small pieces of apple peel were dried in sunlight for 5 days and then in a hot air oven at 60 °C for 48 h. Dried peels were powdered and sieved with a 2 mm mesh size.

The peel extract was prepared by mixing 10 g of apple peel powder in 100 mL of double distilled water and stirring at 75 °C for 2 h [25]. The solution was left for 10 to 15 min to cool and then filtered using Whatman filter paper. While placing the extract on a magnetic stirrer, 3.01 g of iron chloride (FeCl_3_) and 8.61 g of sodium acetate (C_2_H_3_NaO_2_) were added. After mixing for 2 h at 80 °C, the solution was cooled to 25 °C. The solution was centrifuged at 4000 rpm for 15 min, the upper liquid portion was discarded, and the remaining solid part was washed with distilled water two to three times and then kept in a hot air oven at 50 °C for 2 days to dry completely. The final product was ground into a fine powder and preserved at room temperature.

### 2.2. Characterization of Synthesized Nanoparticles

#### 2.2.1. Ultraviolet Visible (UV–Vis) Spectroscopy

A UV–Vis spectrophotometer was used to determine the optical properties of Fe_3_O_4_ NPs. The nanoparticles detected by a UV–Vis spectrophotometer (Lambda 35, Perkin Elmer, USA) between 200 and 800 nm were identified as Fe_3_O_4_ NPs. 

#### 2.2.2. Fourier Transform Infrared Spectroscopy (FT-IR) 

FTIR spectrum was obtained using an Agilent Technologies Cary 630 FTIR, (Fourier transform infrared spectrophotometer) Japan. On the basis of their chemical composition, synthesized nanoparticles were examined. The solution was characterized in the 500–4000 cm^−1^ range by using the KBr pellet method.

#### 2.2.3. X-ray Diffraction (XRD) Analysis

The phase variety and the size of the grains of the created nanoparticles were determined using X-ray diffraction spectroscopy (XRD) (Philips PAN analytical, UK). CUK radiation was used to evaluate the produced nanoparticles at 30 kV of voltage and 20 MA of current, with a scan rate of 0.030/s. The prepared samples’ particle sizes were determined using Scherrer’s equation as follows:$$D\;=\frac{K\lambda }{\beta cos\theta }$$
where D is the size of the crystals, λ is the X-ray wavelength, *θ* is the Braggs angle in radians, *β* is full width at half the maximum of the peak, and *k* is a constant.

#### 2.2.4. Scanning Electron Microscopy (SEM) 

The morphological properties of synthesized apple peel extract-derived Fe_3_O_4_ NPs (JSM-6480 LV, UK) were examined using scanning electron microscopy (SEM). To make the sample conductor, a thin layer of platinum was coated on them. The sample was then examined in the SEM with an accelerating voltage of 20 KV.

### 2.3. Tissue Culture of Turmeric Plant

The turmeric rhizomes were purchased from a nursery in Lahore, Punjab, Pakistan. Clean rhizomes were then buried in wet sand to begin sprouting. The tips of newly sprouting shoots were removed and utilized as explants. To further clean the sprouting rhizomes, they were treated with a fungicide solution containing 1% bavistin for 30 min. The sprouting rhizome buds were removed until the explants’ length was 2–3 cm and the buds were still intact. The removed rhizome segments that enclosed the sprouting buds were subsequently treated for 30 min with a solution of 1% Cabrio, and then they were rinsed with double-distilled water. Then, they were surface sterilized in a laminar flow chamber with 0.2% mercuric chloride (HgCl_2_) for 10 min. The explants were treated with 70% ethanol. Finally, the explants were washed 4 to 5 times with double distilled water. All the other procedures, including final washing with sterile water to remove traces of chloride, cutting, and culturing, were carried out in a laminar flow chamber.

Sprouting rhizome bud explants were cultured in culture tubes using MS [25] culture medium supplemented with 1 mL/L of IAA (indole acetic acid), 8.88 mL/L of BAP (6 benzylaminopurine), and five different concentrations (1, 5, 10, 15, and 20 mg/L) of Fe_3_O_4_ NPs. The cultured tubes were incubated in vitro for four weeks, maintaining the standard culture temperature of 25 °C at 16/8 h photoperiod. The data were collected after 30 days of culturing. The whole experiment was run in triplicate.

### 2.4. Preparation of Turmeric Extract

Samples (control and grown in presence of 15 mg/L of Fe_3_O_4_ NPs) of tissue cultured plantlets of *C. longa* were extracted using an ultrasonic apparatus and ethanol/water (7:3) for 15 min. Each extract was filtered and then evaporated under vacuum in a rotary evaporator at 40 °C. The residue was diluted in a suitable volume of MeOH-water and filtered through a syringe filter (polytetrafluoroethylene, PTFE, 2.5 mm, 0.45 mm) before further analysis through LC-Ms.

### 2.5. Liquid Chromatography–Mass Spectroscopy (LC-MS)

The LC-Ms analysis was carried out by using an Agilent Jet Stream, (Agilent, Santa Clara, CA, USA) with a 996 photodiode assay (PDA) detector together with an automatic liquid chromatographic sampler and an auto injection system hyphenated to a Micromass Quattro Ultima tandem quadrupole mass spectrometer equipped with an electrospray ionization (ESI) source. The separation was accomplished by using a ZORBAX C-18 CA, USA column. The system produced a steady flow of 100 mL/min, and the mobile phase contained 0.1% formic acid and 70% acetonitrile. The injection’s volume was 0.5 µL. The temperature control mode was set at 30 °C. The electrospray interface (ESI) was used for the MS operation mode. A mode 22 multiple syringe pump was used to carry out the analyte infusion experiment. Quantification of the compounds was based on the peak area ratios, carried out by comparing the mean area response of three replicates of extracted samples.

### 2.6. Statistical Analysis

Statistical analyses were performed using the Excel software (version 2019) add-in DSAASTAT 1.101. The experiments were repeated twice, and mean data are presented. The experimental data were analyzed statistically by performing a one-way analysis of variance (ANOVA) and Tukey’s test (*p* = 0.05) to identify significant level differences between treatment means. Data visualization was performed using Excel software, and the standard error of the mean was presented as error bars in the graphical representations.

## 3. Results

### 3.1. Effect of Fe_3_O_4_ NPs on the Regeneration of Turmeric Plant through Tissue Culture

#### 3.1.1. Effect of Fe_3_O_4_ NPs on Callus Induction

Green synthesized Fe_3_O_4_ NPs significantly enhanced rate of callus induction (Figure 1). Increase in concentration of nanoparticles from 1–15 mg/L linearly increased percentage of callus induction. However, further increase in Fe_3_O_4_ NPs to 20 mg/L declined this percentage. The highest Fe_3_O_4_ NPs concentration of 15 mg/L showed an 80% rate of callus induction that was 30% more than the control. At 10 mg/L of Fe_3_O_4_ NPs, the explant showed a satisfactory growth rate with a percentage of 70% in comparison to the control, which showed a 50% rate of callus induction. In the presence of nanoparticles, not only callus induction increased but also the time for callus initiation decreased significantly. Explant in the presence of Fe_3_O_4_ NPs in 15 mg/L was initiated after 28 days in comparison to the control, where initiation was recorded at the 42nd day of inoculation (Table 1).

#### 3.1.2. Effect of Fe_3_O_4_ NPs on In Vitro Shoot Growth of Turmeric

Table 2 shows that among all the concentrations of Fe_3_O_4_ NPs, the MS media supplemented with 15 mg/L showed greater effects on shoot growth by showing the highest percentage of 70% of shoot growth response, an average of 2.5 shoots per explant, 9 cm as the highest shoot height, 4 leaves per plantlet, and 100 mg of fresh weight. Noticeably, the lowest response rate (42%), least number of shoots, and minimum dry weight were recorded when the explants were cultured in MS media without nanoparticles. In general, the MS media supplemented with 15 mg/L of Fe_3_O_4_ NPs showed the most efficient shoot regeneration (Figure 2).

#### 3.1.3. Effect of Fe_3_O_4_ NPs on In Vitro Root Induction

The explant when cultured in MS media supplemented with 1, 5, 10, 15, and 20 mg/L of Fe_3_O_4_ NPs showed an increase rate of root induction. The highest percentage of root induction of 75%, highest number of roots of six roots per plantlet, and highest fresh weight of 51 mg per plantlet were recorded at 15 mg/L concentration as shown in Table 3, while minimal root induction of 45% was shown in control media without nanoparticles. In comparison to the control, the percentage of rooting was 30 and 17% higher in 15 and 10 mg/L of Fe_3_O_4_ NPs. 

### 3.2. Liquid Chromatography–Mass Spectrometry (LC-MS) Analysis of Turmeric

For the determination of curcuminoids in turmeric, a liquid chromatography–mass spectrometry (LC-MS) technique was performed and evaluated as per the guidelines for standard laboratory procedures. A full scan in positive ion modes (range of scan from 200 *m*/*z* to 500 *m*/*z*) was used to determine the analytes. To identify curcumin full scan mass spectra (precursor ion is 369 [M + H^+^]), demethoxycurcumin (precursor ion is 339 [M + H^+^]), bisdemethoxycurcumin (precursor ion is 309 [M + H^+^]), and dihydrocurcumin (precursor ion is 371 [M + H^+^]), 60 V of voltage was used. Collision energies of 15 eV were optimum for the primary production of curcumin at 177 *m*/*z*. The compounds were identified using positive ionization mode by observing the precursor–product mixtures in MRM mode. After optimization, the mass transition for curcumin was *m*/*z* 369–*m*/*z* 177 and had excellent symmetry and significant intensity. The retention time of curcumin was 17.91 min (Table 4). 

The curcuminoids combination revealed relatively lower quantities of dihydrocurcumin. Figure 3 shows a representation of chromatograms of turmeric plant extracts, both of control and grown in the presence of Fe_3_O_4_ NPs. The quantities of curcumin, demethoxycurcumin, bisdemethoxycurcumin, and dihydrocurcumin in the extract of plants grown in media supplemented with Fe_3_O_4_ NPs were found to be 13.73, 8.05, 2.16, and 0.02 mg/g, respectively. The content of curcumin in both extracts was greater than that of the other three compounds. While in the extract of the control plants all the contents of curcumin, demethoxycurcumin, bisdemethoxycurcumin, and dihydrocurcumin were in lower quantity than in plants grown in the presence of Fe_3_O_4_ NPs. 

### 3.3. Characterization of Green Synthesized Fe_3_O_4_ NPs

#### 3.3.1. UV–Vis Spectroscopy

The ultraviolet–visible (UV) absorption spectrum of the green synthesized Fe_3_O_4_ NPs is shown in Figure 4 as the wavelength range of 250–800 nm. The major absorption bands of Fe_3_O_4_ NPs were identified at 290 nm. Upon exposure to light, Fe_3_O_4_ NPs absorb light, which may result in the excitation of electrons from the valence to the conduction band.

#### 3.3.2. Fourier Transformed Infrared Spectroscopy (FTIR)

Fourier transformed infrared spectroscopy revealed detailed data about the profile of infrared spectra absorption on the Fe_3_O_4_ NPs sample. FTIR analysis for the wavelength range of 500–4000 cm^−1^ was used for the characterization of Fe_3_O_4_ NPs. The data plot displays the infrared light wavelength as distinct absorption peaks at specific wavenumbers that are caused by the vibration of particular functional groups. The stretching vibration of Fe-O functional groups takes place at the absorption of wavenumbers between 525 cm^−1^ to 571 cm^−1^ (Figure 5). The presence of the H-O-H bending vibration can be noted in the wavenumber range of 1327–1665 cm^−1^, whereas the OH group vibrates stretching between the wavenumber intervals from 3229 cm^−1^ to 3519 cm^−1^.

#### 3.3.3. X-ray Diffraction (XRD)

The crystalline structure and phase composition of Fe_3_O_4_ NPs synthesized using apple peel extract were revealed by X-ray diffraction (XRD) analysis. The XRD pattern exhibits unique diffraction peaks at certain angles, which correspond to the Fe_3_O_4_ NPs’ crystal planes. The locations and strengths of these points allow the crystal structure to be identified. The creation of magnetite Fe_3_O_4_ NPs is suggested by the existence of distinctive peaks at 30.1° 35.6°, 57.2°, and 62.6° [23]. The relative strengths of the peaks in diffraction reveal the amount of various crystal phases inside Fe_3_O_4_ NPs. To determine the phase composition, the peak intensity ratios may be compared to standardized reference data. The XRD pattern shows that the magnetite phase predominates in the Fe_3_O_4_ NPs synthesized with apple peel extract.

The lack of further diffraction peaks, or the presence of just faint, minor peaks, confirms the Fe_3_O_4_ NPs phase purity. Any unusual peaks might indicate the presence of contaminants or subsequent phases. The XRD pattern of Fe_3_O_4_ NPs shows relatively high phase purity, validating the effective production of magnetite Fe_3_O_4_ NPs as shown in Figure 6.

#### 3.3.4. Scanning Electron Microscopy (SEM)

Fe_3_O_4_ NPs made from the apple peel extract were examined using SEM (scanning electron microscopy), which revealed information on the shape, size distribution, and surface properties of the nanoparticles. An explanation of how to read Fe_3_O_4_ NPs generated from extracts of apple peel from SEM pictures is given below (Figure 7). The size of the nanoparticles ranged from those that are smaller (10–20 nanometers) to those that are bigger (100–200 nanometers).

## 4. Discussion

In recent times, nanoparticles with the ability to improve plant physiological systems have been synthesized for their application in agriculture and biotechnology. They have been presented as mineral substances, especially important micronutrients needed for plant defense, as stimuli of resistance, and as agents that reduce the occurrence of plant pathogens [26]. In the present study, Fe_3_O_4_ NPs were green synthesized as iron plays an important function in many plant physiological systems, i.e., redox reactions and the formation of chlorophyll. Iron is an essential component for the metabolism and development of plants, and its insufficient amount may cause a prevalent nutritional disease in several plants, leading to reduced production and output [27]. These characteristics show the potential use of nanoparticles in tissue culture regeneration of plants to enhance their growth.

In the current study, the Fe_3_O_4_ NPs showed a great effect on callus induction and plant growth. In comparison to the control, callus induction was 30% higher in those that were grown in 15 mg/L of synthesized nanoparticles. Being small in size, nanoparticles are easily taken up by the plants through various reported mechanisms. The transportation within plants can be symplastic or apoplastic [28]. It is known that rapidly growing cells within the explant show extensive multiplication, and therefore there is a need for active synthesis of DNA. This great prerequisite for nucleic acid synthesis supports augmented manufacturing of nucleoside triphosphate (NTP), an initial substrate for synthesis of nucleic acids. This increased NTP synthesis increases pH level within cells, and Fe_3_O_4_ NPs are chemically stable in alkaline pH, hence available for rapid plant growth. It is also known that because of their large surface area, nanoparticles can attach to organic chemicals or to carrier proteins present in the cell membranes that help them in migration [29]. Once within a cell, these nanoparticles may have an impact on genetic regulation [30], thus affecting physiological and biochemical processes within a cell. This may lead to enhanced callus induction, growth, and disease resistance. A similar study conducted by [31] confirmed the positive role of iron nanoparticles on plantlet regeneration of *Panax vietnamensis*. However, they reported adverse effects on growth when nanoparticles were added in a concentration of 11.2 mg/L, whereas in the current study, a concentration of 15 mg/L was found to support both callus induction and growth in turmeric. Beside differences in plants, this may be due to differences in synthesis of nanoparticles used. Ref. [32] used chemically synthesized iron nanoparticles [33], whereas the present investigation was carried out with green synthesized iron nanoparticles. This also confirms that green synthesis protocols can reduce toxicity of metal and metal oxide nanoparticles. Further increase in concentration of Fe_3_O_4_ NPs to 20 mg/L incurred adverse effects on micropropagated plantlets. Ref. [34] reported that iron oxide nanoparticles at high concentrations can decrease cell viability. High iron content can trigger the formation of reactive oxygen species (ROS), resulting in damage to plant cells. According to [35], iron can transmit electrons to oxygen to form H_2_O_2_, thus extraordinary reactive OH radicals accumulate, causing degradation of cell wall polysaccharides [36]. The positive effects of treatment with Fe_3_O_4_ NPs on K accumulation could be due to iron-dependent activation of NADPH oxidases, since the activity of these enzymes is essential for controlling intracellular K+ homeostasis via ROS-gated ion channels [26].

Growth of turmeric plantlets was also found to be increased in the presence of Fe_3_O_4_ NPs in concentrations between 10–20 mg/L. The increased growth indicates that iron, as a micronutrient, has supported growth-related physiological processes [14,19]. It is also reported that the effects of nanoparticles in plants depend on concentrations, species of plants, and exposed timing [37]. Similar findings were presented by [22] in the chickpea plant, which showed an increase in shoot growth. Therefore, it is reasonable to believe that Fe_3_O_4_ NPs affected plant metabolism, and they may also cling to plant roots and alter the morphology and physiology of turmeric plantlets [26]. 

An increase in callus induction can indirectly be correlated to the antimicrobial potential of iron nanoparticles [38,39,40]. Microbial contamination at different stages of tissue culture is a main reason for low propagation rates. Ref. [41] explained the role of nanoparticles as elicitors in tissue culture. It is well documented that metal and metal oxide nanoparticles can activate defense in plant cells by activating enzymes such as superoxide dismutase, catalase, peroxidases, and polyphenol oxidases [42]. 

Plant growth regulators (PGR), including auxins, cytokinins, gibberellins, abscisic acid, and ethylene, have key roles in plant growth and development. The role of auxin is most important in initiation of the apical meristem. Cytokinin is involved in germination, meristematic functions, and leaf senescence, whereas abscisic acid controls germination. Ref. [43] in a study explained the role of biosynthesized AgNPs on hormones and hence on regenerating rice calli. They reported downregulation of PGR genes in nanoparticle-treated cells. The lowest expression was recorded in cells treated with 10 mg/L of nanoparticles that showed a 20% increase in callus initiation and induction in rice. In this investigation, beside callus induction, callus initiation was also improved with various doses of Fe_3_O_4_ NPs. Moreover, 15 mg/L nano-supplemented media showed callus initiation 14 days earlier than the control.

The green synthesis of Fe_3_O_4_ NPs was validated with a UV–Vis absorption band at 236 nm and stretching vibrations of Fe-O at the absorption of wave numbers between 525 and 571 cm^−1^. These optical characteristics were found to be closely similar to the previous investigations as of [44,45]. XRD revealed diffraction peaks corresponding to the cubic phase of metallic iron, resulting in a highly crystalline structure. The size of the synthesized nanoparticles was as small as 10–20 nm, making them easier to uptake and transport by the cells. However, larger particles of 100 nm and more were also present. It is known that small-sized nanoparticles can easily cling to the cell membranes and create small pores that then help relatively larger nanoparticles to travel in a system. 

Plants are a significant source of potentially valuable active components, including novel antimicrobial agents [46]. Curcuminoids are vital compounds of turmeric. Synthesized Fe_3_O_4_ NPs not only supported micropropagation of turmeric but also increased curcuminoids in the whole plant. The findings are consistent with some other workers [47,48,49] reporting an enhanced production of curcuminoids after application of ZnO nanoparticles on *Curcuma longa*. Hence the study confirms that the green synthesized Fe_3_O_4_ NPs, when used in concentrations of 15 mg/L, can positively affect micropropagation as well as curcuminoids synthesis in turmeric [50].

## 5. Conclusions

This study reveals that biofabricated Fe_3_O_4_ NPs can be a more biologically compatible, efficient, eco-friendly, and reasonable method that may be employed effectively in investigations using plant tissue culture. Fe_3_O_4_ NPs 15 mg/L showed 80% increase in callus induction, 70% enhancement in shoot growth, and 75% increase in root induction in the turmeric plantlets. The treatment also supported 0.3 times more production of curcuminoids. Fast callus initiation can help us save time and valuable resources and thus can enhance the efficacy of micropropagation of turmeric.

## Figures and Tables

**Figure 1 plants-13-01819-f001:**
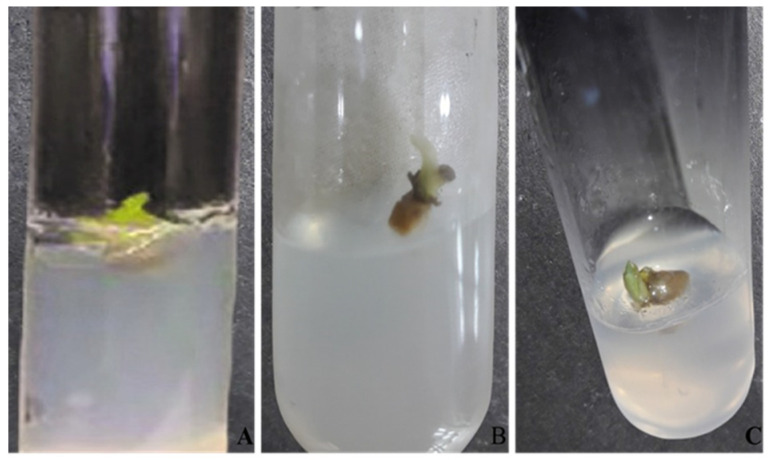
Comparison of callus induction on MS media supplemented with different concentrations of Fe_3_O_4_ NPs after 28 days of inoculation. (**A**) Simple MS media (control); (**B**) MS media supplemented with 10 mg/L; (**C**) MS media supplemented with 15 mg/L.

**Figure 2 plants-13-01819-f002:**
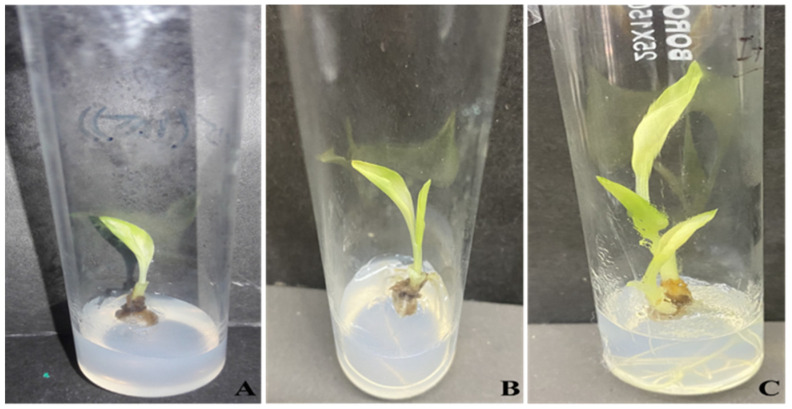
Effects of Fe_3_O_4_ NPs on in vitro growth of turmeric. (**A**) Simple MS media (control); (**B**) MS media supplemented with 10 mg/L; (**C**) MS media supplemented with 15 mg/L.

**Figure 3 plants-13-01819-f003:**
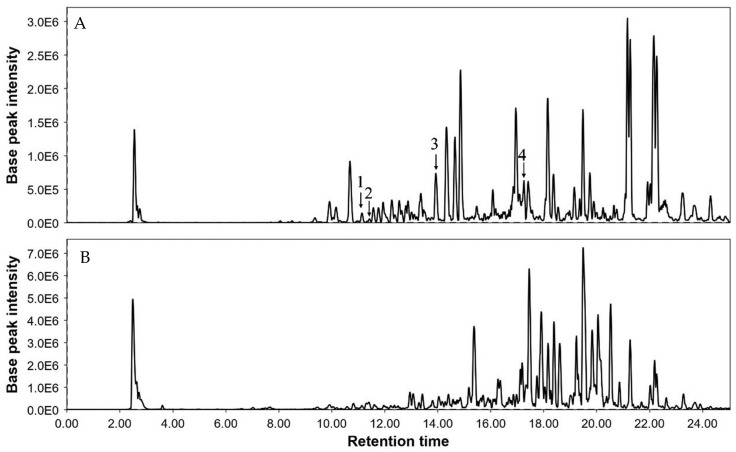
Representative chromatograms of turmeric extract of plants grown in presence of Fe_3_O_4_ NPs (**A**) and control plants (**B**). Where 1 = dihydrocurcumin, 2 = bisdemethoxycurcumin, 3 = demethoxycurcumin, and 4 = curcumin.

**Figure 4 plants-13-01819-f004:**
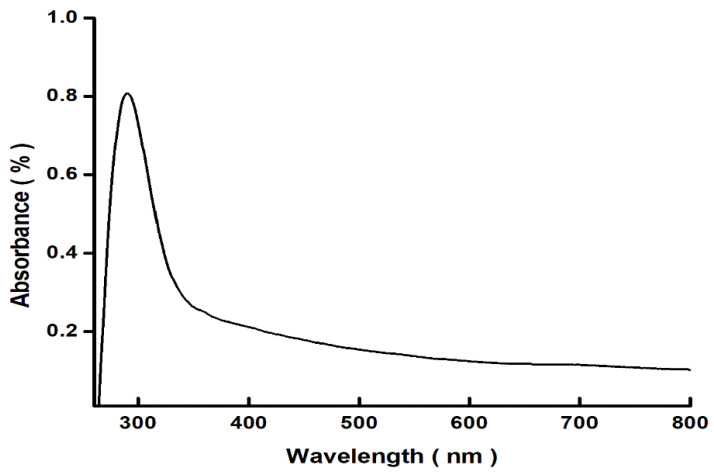
A graphic illustration of the UV–Vis spectrophotometric study displaying the absorbance spectra of green Fe_3_O_4_ NPs that were synthesized using apple peel extract.

**Figure 5 plants-13-01819-f005:**
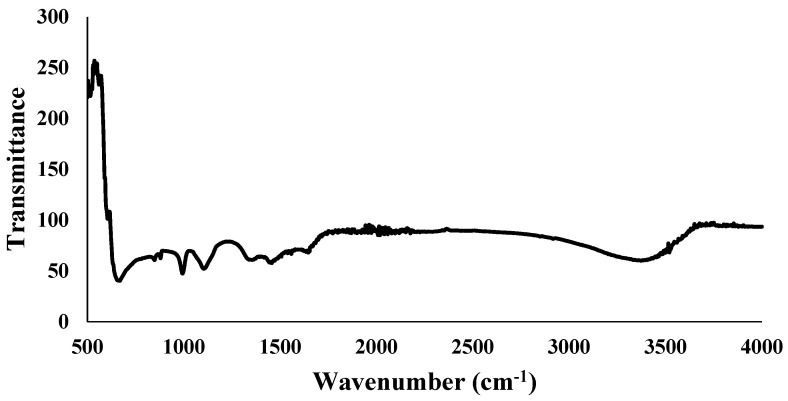
Fourier transform infrared (FTIR) spectra of green synthesized Fe_3_O_4_ NPs using apple peel extract.

**Figure 6 plants-13-01819-f006:**
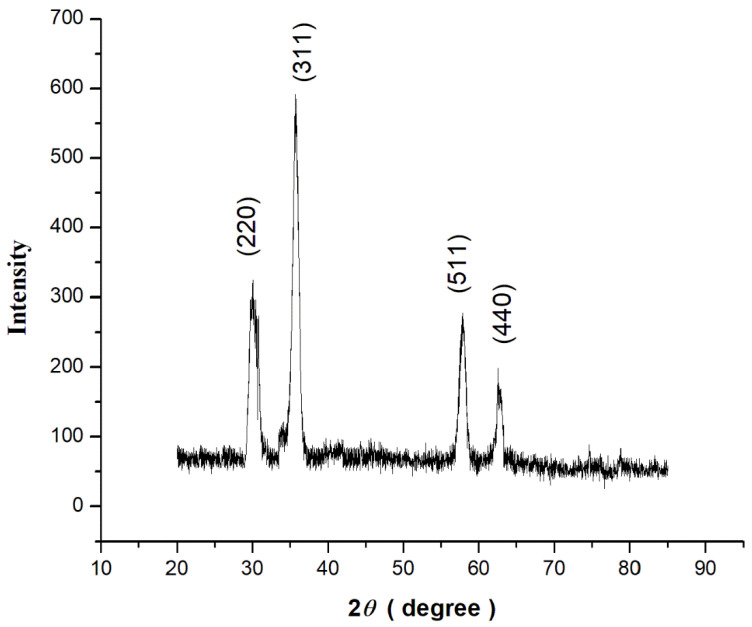
X-ray diffraction (XRD) pattern (before extension of the peaks) of green synthesized Fe_3_O_4_ NPs using apple peel extract.

**Figure 7 plants-13-01819-f007:**
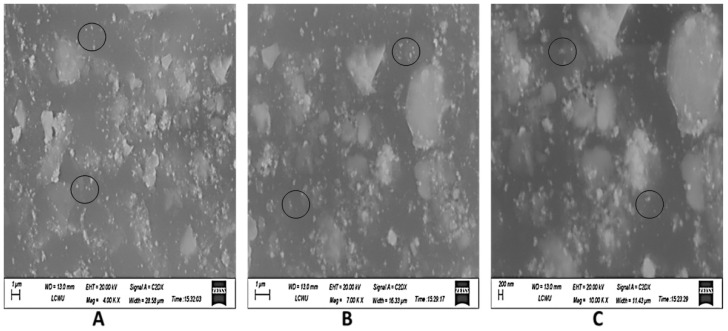
Scanning electron microscopy (SEM) image of green synthesized Fe_3_O_4_ NPs. (**A**) = 4×, (**B**) = 7×, and (**C**) = 10×.

**Table 1 plants-13-01819-t001:** Effect of different concentrations of Fe_3_O_4_ NPs on callus induction.

Treatment	Callus Initiation (Day)	Callus Induction (%)	Proliferation Rate
No Fe_3_O_4_ NPs (Control)	42	50	++
Fe_3_O_4_ NPs (1 mg/L)	42	52	++
Fe_3_O_4_ NPs (5 mg/L)	39	55	++
Fe_3_O_4_ NPs (10 mg/L)	33	70	+++
Fe_3_O_4_ NPs (15 mg/L)	28	80	++++
Fe_3_O_4_ NPs (20 mg/L)	41	40	+

++++: Excellent; +++: Good; ++: Satisfactory; +: Poor.

**Table 2 plants-13-01819-t002:** Effect of Fe_3_O_4_ NPs on in vitro shoot growth of turmeric.

Treatments	Percent Response (%)	Shoot Height (cm)	No. of Leaves per Plantlet	Shoot Fresh Weight (mg)
No Fe_3_O_4_ NPs (Control)	42.3 ± 0.8 ^b^	3.4 ± 0.4 ^b^	1.3 ± 0.3 ^b^	51.3 ± 0.6 ^b^
Fe_3_O_4_ NPs (1 mg/L)	44.0 ± 1.0 ^b^	3.7 ± 0.2 ^b^	1.3 ± 0.3 ^b^	52.6 ± 0.7 ^b^
Fe_3_O_4_ NPs (5 mg/L)	46.0 ± 0.5 ^b^	4.5 ± 0.5 ^bc^	2.0 ± 0.5 ^b^	57.1 ± 0.8 ^c^
Fe_3_O_4_ NPs (10 mg/L)	60.0 ± 1.1 ^c^	6.2 ± 0.7 ^c^	3.0 ± 1.0 ^c^	74.1 ± 1.0 ^d^
Fe_3_O_4_ NPs (15 mg/L)	70.0 ± 0.5 ^d^	9.0 ± 0.7 ^d^	5.6 ± 0.8 ^d^	100.3 ± 0.1 ^e^
Fe_3_O_4_ NPs (20 mg/L)	33.2 ± 0.5 ^a^	2.1 ± 0.4 ^a^	1.1 ± 0.4 ^a^	40.2 ± 0.7 ^a^

Data represent mean ± standard error. Different letters denote statistically significant differences between treatments as evaluated by the ANOVA and Tukey’s multiple range test at the *p* = 0.05 level.

**Table 3 plants-13-01819-t003:** Effect of Fe_3_O_4_ NPs on root induction of turmeric.

Treatments	Percent Rooting (%)	No of Roots	Root Fresh Weight (mg)
No Fe_3_O_4_ NPs (Control)	45.3 ± 1.8 ^a^	1.3 ± 0.3 ^a^	27.0 ± 1.7 ^a^
Fe_3_O_4_ NPs (1 mg/L)	47.6 ± 0.3 ^a^	1.6 ± 0.3 ^a^	29.2 ± 0.5 ^a^
Fe_3_O_4_ NPs (5 mg/L)	49.3 ± 0.8 ^a^	2.3 ± 0.6 ^ab^	30.2 ± 0.6 ^a^
Fe_3_O_4_ NPs (10 mg/L)	62.0 ± 2.1 ^b^	4.0 ± 0.5 ^b^	38.2 ± 1.9 ^b^
Fe_3_O_4_ NPs (15 mg/L)	75.0 ± 2.1 ^c^	7.0 ± 0.5 ^c^	51.1 ± 2.2 ^c^
Fe_3_O_4_ NPs (20 mg/L)	43.3 ± 0.9 ^a^	1.1 ± 0.2 ^a^	25.0 ± 2.1 ^a^

Data represent mean ± standard error. Different letters denote statistically significant differences between treatments as evaluated by the ANOVA and Tukey’s multiple range test at the *p* = 0.05 level.

**Table 4 plants-13-01819-t004:** Detection of curcuminoids and their metabolites.

Compound	MW	RT	Formula	*m*/*z*	Concentration mg/g	
					With NPs	Without NPs
Curcumin	368	17.91	C_12_H_20_O_6_	369, 285, 177	13.73	10.35
Demethoxycurcumin	338	14.10	C_20_H_18_O_5_	339, 177, 147	8.05	6.49
Bisdemethoxycurcumin	308	10.75	C_19_H_16_O_4_	309, 225, 147	2.16	1.13
Dihydrocurcumin	370	10.50	C_12_H_22_O_6_	371, 137, 177	0.02	0.01

## Data Availability

All data generated or analyzed during this study are included in this published article.

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
