# Peer review of "Nano-Integrated Plant Tissue Culture to Increase the Rate of Callus Induction, Growth, and Curcuminoid Production in *Curcuma longa"

_plants, 2024, doi:10.3390/plants13131819_

Round 1
Reviewer 1 Report (Previous Reviewer 1)
Comments and Suggestions for Authors
This is a research manuscript about efficacy of micropropagation in turmeric plants using iron oxide nanoparticles. This manuscript, in its structure and content, fits with the subject matter of this journal. The content of the manuscript is informative in both scientific and practical terms.
Since one of the objectives of the study is the saving of time and resources during micropropagation, some justification and comparison in comparison with the usual classical process would be needed, so that it would be possible to evaluate the aspects of time and costs.
The manuscript is based on 49 references from year 1962 to 2023. A number of literature sources (~60%, 29/49) are older than 5 years, so I suggest updating the bibliography with newer sources (as much as it is possible for the authors) with the latest scientific discoveries in the field.
Reference source #26 is mixed with two references. The citation in the text should be reviewed to see if it corresponds to the reference numbers.
Reference source #42 does not have year provided.
The pictures and figures in the manuscript illustrate the topic well and fit well into the context.
The English language of the manuscript is good, no significant proofreading errors have been observed.
The SEM images are quite unclear, maybe it would be possible to make them more specific (e.g. by circling a few particles). Because now it seems that the majority of that particle is much larger than what is written in the manuscript. And do they differ in size due to the formation of conglomerates or for other reasons (as you think)?
The conclusions are very general, it supports the main idea of the research, but it would be good to make them a little more specific.
Author Response
please see Attachment.

Reviewer 2 Report (New Reviewer)
Comments and Suggestions for Authors
I went through this manuscript few times before coming to this conclusion. The title says 'Nano-integrated plant tissue culture to increase the rate of callus induction, growth, and curcuminoids production in Curcuma longa'. However, there is no information on callus induction or curcuminoid (which is different from curcumin).
The major issue with the manuscript is- it largely deals with synthesis of nanoparticles and characterize them. The data and the results on the effects of turmeric is highly questionable. Further they say callus in the title, but there is no mention about callus in the methodology or results. The measurement of curcumin is irrelevant to this study. Measuring curcumin is aerial parts of turmeric is meaningless as they are not consumed or commercially used otherwise.
Ideally if they just describe the nanoparticle synthesis and characterization as the primary work and add some better data on its effect on turmeric growth in vitro- it can be publishable in a Nanoparticle related journal but not Plants.
I am sorry I could not recommend this manuscript for publication in Plants; however, I encourage the authors to take out the ancillary part on in vitro culture and just describe the NP synthesis and characterization alone in depth and submit it to a Nanotechnology related journal where the chances of getting published is much higher.
Comments on the Quality of English LanguageBarring some minor issues I don't see anything major with English.
Author Response
please see Attachment.

Reviewer 3 Report (New Reviewer)
Comments and Suggestions for Authors
The manuscript described the synthesis and primary characterization of Fe3O4 NPs using apple peel extract and its effects on enhancing callogenesis, shoot regeneration patterns, root induction and curcuminoid contents in turmeric plants. The results are some of interest. Some questions should be further clarified. What’s the effects and the mechanism of apple peel extract during Fe3O4 NPs synthesis reaction? Why Fe3O4 NPs show major absorption bands at 236 nm? The unit of wavenumber in figure 5 should be clarified. No obvious diffraction peaks were observed in XRD analysis. The SEM images were not clear enough.
Round 2
Reviewer 1 Report (Previous Reviewer 1)
Comments and Suggestions for Authors
The comments and suggestions made have been taken into account, the manuscript has been improved, I have no further comments, and I consider the manuscript suitable for publication.
Reviewer 2 Report (New Reviewer)
Comments and Suggestions for Authors
I am sorry but I am not in agreement with your explanations. If the other reviewers agree with your argument, then I will leave it to the editor. Still the manuscript revolves mostly around synthesis and characterization of NPs and the results shown are not convincing enough for the increase in curcuminoids/curcumin.
Comments on the Quality of English LanguageEnglish is fine. Nothing really major to worry about.
This manuscript is a resubmission of an earlier submission. The following is a list of the peer review reports and author responses from that submission.
Round 1
Reviewer 1 Report
Comments and Suggestions for Authors
This is a research manuscript about synthesis of Fe nanoparticles to increase the efcacy of micropropagation. This manuscript, in its structure and content, fits with the subject matter of this juournal.
The content of the manuscript is quite interesting and informative in both scientific and practical terms.
The literature reviewed in the manuscript covers the period from 1962 to 2023. About 65% of literature sources are older than 5-10 years, so I suggest updating the bibliography with newer sources (as much as it is possible for the authors) with the latest scientific discoveries in the field.
The figures and tables in the manuscript illustrate the topic well and fit well into the context, however I would really like to see the comparative spectra in figures 4 and 5, where the extracts with and without NPs are compared (similar to the one presented with the chromatograms).
The conclusions are very general, but supports the main idea of the study.
Reviewer 2 Report
Comments and Suggestions for Authors
Here is my report about the ms “Nano-integrated plant tissue culture to increase the rate of callus induction, growth, and curcuminoids production in Curcuma longa”
After reading the ms, I feel that the ms is not suitable for publication in high quality journals.
The writing process was superficial, particularly the discussion section.
L105-106: The peel extract was prepared by mixing 10g of apple peel powder in 100 mL of double distilled water and stirring at 75 ºC for 2 hours. Why authors used these material as dry instead of using the fresh material of apple?
L176 Statistical analysis section should be rewritten in detail showing the different factors.
The statistical analysis for the obtained results is not appreciated while presenting the data in the Results section.
Figures, particularly Figure 7 are very low quality, it seems to copy past from somewhere!
The discussion section is written like an essay. The authors should make it a real discussion by discussing their results based on their treatments and discussing these findings with others using how and why tools! Please focus on the mechanisms of bio-fabricated Fe3O4 NPs and why the can be a more biologically compatible, efficient, eco-friendly and reasonable method. I am sorry, how Fe3O4 NPs can be reasonable method? Do you think so?
Comments on the Quality of English Language
Some sentences need moderate editing.
Reviewer 3 Report
Comments and Suggestions for Authors
The work synthesizes 'green' NPs of iron for use in plant micropropagation studies involving Curcuma. The study lacks a control that identifies the nanoparticle synthesis as the critical feature. In particular, what happens when FeO3 is introduced as a simple reagent in non-nanoparticle form? The results are moderately definitive from this reviewers perspective. For example, in Fig. 1 it is suggested by the authors that significant differences in callus induction is shown but this is not clear and not aided by any legend or figure caption. The proliferation rate in Table 1 seems to be a subjective scale and no particular metrics are identified. Tables 2 and 3 show values with superscript a-e, but this is not clearly identified as any post-ANOVA comparison. The spectra in Fig. 4 is unremarkable and is likely to show scattering if indeed nanoparticles are present. Fig. 6 is difficult is understand as nothing seems to be identified at the designated locations. In Fig. 7 the scale shows a size bar of 200 nm and 1 micron - the sizes shown seem to be well above those values, but the picture is not well focused. The nano-nature of the particulates is very much in doubt.